# Coumarin-Chalcone Hybrids as Inhibitors of MAO-B: Biological Activity and In Silico Studies

**DOI:** 10.3390/molecules26092430

**Published:** 2021-04-22

**Authors:** Guillermo Moya-Alvarado, Osvaldo Yañez, Nicole Morales, Angélica González-González, Carlos Areche, Marco Tulio Núñez, Angélica Fierro, Olimpo García-Beltrán

**Affiliations:** 1Biology Department, Johns Hopkins University, Baltimore, MD 21218, USA; gmoya@bio.puc.cl; 2Center of New Drugs for Hypertension (CENDHY), Santiago 8330015, Chile; osvyanezosses@gmail.com; 3Computational and Theoretical Chemistry Group, Departamento de Ciencias Químicas, Facultad de Ciencias Exactas, Universidad Andres Bello, República 498, Santiago 7550196, Chile; 4Department of Physiology, Faculty of Biological Sciences, Pontificia Universidad Católica de Chile, Santiago 8331150, Chile; nlmorales@uc.cl; 5Laboratorio de Interacciones Insecto-Planta, Instituto de Ciencias Biológicas, Universidad de Talca, Casilla 747, Talca 3460000, Chile; angelica.gonzalez@utalca.cl; 6Department of Chemistry, Faculty of Sciences, Universidad de Chile, Las Palmeras 3425, Nuñoa, Santiago 7800024, Chile; areche@uchile.cl; 7Biology Department, Faculty of Sciences, Universidad de Chile, Santiago 7800024, Chile; mnunez@uchile.cl; 8Department of Organic Chemistry, Faculty of Chemistry, Pontificia Universidad Católica de Chile, Casilla 306, Santiago 6094411, Chile; 9Centro Integrativo de Biología y Química Aplicada (CIBQA), Universidad Bernardo O’Higgins, General Gana 1702, Santiago 8370854, Chile; 10Facultad de Ciencias Naturales y Matemáticas, Universidad de Ibagué, Carrera 22 Calle 67, Ibagué 730002, Colombia

**Keywords:** chalcocoumarin, MAO-B, molecular dynamics, in silico studies, neurodegenerative diseases

## Abstract

Fourteen coumarin-derived compounds modified at the C3 carbon of coumarin with an α,β-unsaturated ketone were synthesized. These compounds may be designated as chalcocoumarins (3-cinnamoyl-2*H*-chromen-2-ones). Both chalcones and coumarins are recognized scaffolds in medicinal chemistry, showing diverse biological and pharmacological properties among which neuroprotective activities and multiple enzyme inhibition, including mitochondrial enzyme systems, stand out. The evaluation of monoamine oxidase B (MAO-B) inhibitors has aroused considerable interest as therapeutic agents for neurodegenerative diseases such as Parkinson’s. Of the fourteen chalcocumarins evaluated here against MAO-B, **ChC4** showed the strongest activity in vitro, with IC_50_ = 0.76 ± 0.08 µM. Computational docking, molecular dynamics and MM/GBSA studies, confirm that **ChC4** binds very stably to the active rMAO-B site, explaining the experimental inhibition data.

## 1. Introduction

Coumarins (α-benzopyrones, 2*H*-chromen-2-ones) are a large family of compounds, of natural and synthetic origin, that show numerous biological [1,2,3,4,5,6] and medicinal chemistry activities, such as anticoagulant, anticancer, antioxidant, antiviral, anti-diabetic, anti-inflammatory, antibacterial, antifungal and anti-neurodegerative properties [7,8,9], among which recent studies have paid special attention to enzyme inhibition. With regard to monoamine oxidase (MAO) inhibition, recent findings have revealed that MAO affinity and selectivity can be efficiently modulated by appropriate substitutions on the coumarin ring system [1,10,11,12,13].

MAOs (EC 1.4.3.4) are flavoproteins located in the outer mitochondrial membrane and involved in the oxidative deamination of endogenous and exogenous monoamines using oxygen (O_2_) as electron acceptor. In humans they exist in two isoforms called MAO-A and MAO-B. The high resolution crystal structures of both human isoforms A and B (hMAO) rat MAO-A (rMAO-A) have made it possible to analyze binding modes of ligands inside these macromolecules [14]. While the active site is formed by the common FAD cofactor and similar amino acid residues in the different forms, these are distinguished by their selectivity for substrates and inhibitors [15]. Thus, serotonin and noradrenaline are substrates of MAO-A which is selectively inhibited by clorgyline, while MAO-B oxidizes β-phenylethylamines and benzylamines and is selectively inhibited by l-deprenyl. MAO genes are expressed in various tissues. However, in the brain, although both isoforms are widely distributed, MAO-B is expressed in high concentrations in the hypothalamus, striatum, globus pallidus and thalamus, and mainly in serotonergic cells while the A isoform is rather evenly distributed, mainly in the cortex, and in nuclei containing preferably catecholaminergic and glial cells [16,17,18,19,20,21].

Although knowledge about MAO inhibition by compounds containing coumarin scaffolds is scarce, publications of articles describing new inhibitors of this class of compounds are increasing. The variety of substitutions on the coumarin ring provide insight into the influence on the activity-structure relationship. Among the most reported modifications of the coumarin ring with MAO activity are on C3 and the steric effect of the substituent appears to be important in modulating MAO-B inhibitory activity [11]. In addition, it has been reported that the introduction of various substituents at the *para* position of the 3-phenyl ring is a good strategy for improving the desired MAO-B inhibitory activity [22] and when the 3-phenyl skeleton is replaced by a 3-benzoyl group, the activity is strongly diminished [20]. It has also been observed that coumarins substituted with 3-indolyl and 3-thiophenyl shows greatest selective inhibition was against MAO-B [11,23,24].

In this work a merger of the coumarin scaffold and a 3-cinnamyl group led to new hybrid (chalcocoumarin) derivatives (Scheme 1) that preserve structural characteristics of compounds with the ability to interact with MAO. The synthetic strategy chosen allowed a large variety of substituents on the cinnamyl benzene ring to be accessed using different readily accessible benzaldehydes. Thus, the quantity and/or type of interactions with the enzyme were explored involving some bulky groups to determine their possible contribution to the biological activity as MAOIs. Our new compounds were screened versus both MAO isoforms, and in silico studies were carried out to rationalize their main interactions in the MAO active cavity. The computational biochemistry tools were used considering the geometrical restrictions and most probable positions in the formation of the ligand-receptor complex. The chalcocoumarin molecules were subjected to theoretical studies in which binding energies were estimated using docking and MM/GBSA analysis. In addition, physicochemical parameters that are responsible for governing the pharmacokinetic properties of drug molecules were determined.

In the present study, a series of coumarin-chalcone hybrid compounds were synthesized and tested on the 2 MAO isoforms. The activity shown was selective for MAO-B and in particular, compound **ChC4** showed the highest inhibitory activity on rMAO-B at submicromolar concentrations. The results obtained will be useful to understand the mode of inhibition of chalcocoumarins against rMAO-B, and to help predict the activities of these new inhibitors that could be promising as therapeutics to treat neurodegenerative diseases such as Parkinson’s disease.

## 2. Results and Discussion

### 2.1. Chemistry

The route employed to synthesize the compounds is summarized in Scheme 1. The compounds were obtained starting from resorcinol (**1**), which was formylated using the Vilsmaier-Haack reaction [25]. Knoevenagel condensation of the aldehyde intermediate with ethyl acetoacetate afforded hydroxycoumarin **2**. The 7-hydroxycoumarin obtained was methylated using a Williamson reaction using methyl sulfate as methylating agent, obtaining the compound **3**, finally the compounds derived the 3-cinnamoyl-2H-chromen-2-one (**ChC1**–**ChC14**) (Table 1) were prepared in moderate yields (25–47%, unoptimized) by Claisen-Schmidt condensation with the respective aldehyde (Appendix A) [26]. Coumarin-chalcone hybrids have been studied and are currently still being synthesized for various uses and their spectroscopy is well known, however, we will detail some signals that are key to their identification. The ^1^H-NMR spectra of the compounds **ChC1**–**ChC14** present very similar chemical shift patterns with a particular signal that identifies this type of molecules, the neighboring vinyl protons of the α,β-unsaturated ketone appear at very close low field from the aromatic proton region. These protons present signals corresponding to two doublets with variable δ between 8.5 and 7.0 and with J_ab_ = 16 Hz on average. this high constant corresponds to a trans isomer [27,28,29]. As for the ^13^C-NMR spectrum, we will mention typical signals such as carbonyl shifts. first of all, we will detail that the carbon of the α,β-unsaturated ketone has a δ 190–180 ppm and carbonyl carbons of α-pyrone δ 165–155 pmm on average [27,28,29], the compounds were characterized by ^1^H and ^13^C NMR (Appendix A).

### 2.2. Biological Analysis in Rat MAO

Fourteen derivatives differing in the substitution pattern of the cinnamyl benzene ring were studied, these compounds were tested on rat MAO-A and B to determine their inhibitory activity MAO. A general screening was carried out at 10 µM finding moderate activity for some of the compounds against rMAO-B but none against rMAO-A. Thus, five molecules were identified as possibly selective IMAO-B.

**ChC4**, **ChC5**, **ChC6**, **ChC9** and **ChC11** in MAO-B exhibited micromolar or submicromolar in vitro potencies, all below 10 µM (Table 2). Out of these **ChC4**, substituted with a hydroxyl group on the meta position of the variable ring, displayed the highest rMAO-B inhibitory activity (IC_50_ = 0.76 µM). Interestingly, changing the position of the hydroxyl group from meta to ortho or para (**ChC2** and **ChC10** respectively) led to loss of the inhibitory activity. An approximately 12-fold lower IMAO activity was observed when the hydroxyl group (in **ChC4**) was methylated (**ChC5**). This might be attributed to steric hindrance and/or to the loss of hydrogen bonding donor quality which could be crucial for some interaction in the binding site. Moving the methoxyl group from the meta to the para position (**ChC6** vs. **ChC5**) slightly increased potency.

Replacing the methoxy group of **ChC6** with a bulkier, less electronegative and more polarisable methylthio group (**ChC11**) only produced **ChC6** is less potent than **ChC11**. The second most potent molecule was **ChC9**, with a methylenedioxy group bridging the meta and para carbons. The methylenedioxy group increases the rigidity of the molecule, possibly stabilizing the complex protein-ligand interaction. The same effect, extending the rigidity, has been observed, on other derivatives as IMAO [30,31].

### 2.3. Molecular Docking and Ligand Efficiency Analysis

To analyze the changes in potency of the coumarin-chalcone hybrids, docking studies were carried out using the crystal structure of rMAO-A and the homology model of rMAO-B (Appendix A), analyzing the possibility that each one of them has to form a stable complex with each of the 14 molecules synthesized by us. Table 3 shows that most of these molecules present better interactions with the rMAO-B binding site, since the corresponding energies are at least 2.5 kcal mol more negative in all but one of the cases. This difference could be due to the substitutes present in the molecules. The results of the molecular docking experiments showed more favorable interactions (more negative ∆*E_binding_*) for the complexes in rMAO-B than in rMAO-A, with average values around −9.26 kcal·mol^−1^ vs. −6.57 kcal·mol^−1^ respectively) which are in accord with the experimental data for the whole series. In the rMAO-B, although no major differences were observed in the binding modes of the active compounds, subtle energy differences were identified. The results of this molecular docking study point to strong interactions of the chalcocoumarins in the binding pocket of rMAO-B, but considerably weaker in rMAO-A.

When analyzing the docking results for rMAO-B from the conformational viewpoint, it is necessary to consider the residues that constitute the substrate-binding site of rMAO-B, which is composed of the FAD cofactor, two flanking residues, Tyr398 and Tyr435, that form an “aromatic box”, and a number of others, particularly Cys172, Tyr326, Met341, Ser200 Gln206 and Thr314 [32,33]. The results show that all the chalcocoumarins settle in the active site of rMAO-B (Appendix A), with the benzene ring of the coumarin moiety close to the FAD, more specifically the central N-5, at a distance of about 4.0 Å. The benzene ring of the cinnamyl moiety extends into the generally hydrophobic entrance cavity adjoining the substrate-binding site. The mere length of the ChC molecules indicates that to bind in the active site of MAO-A the latter must undergo a rearrangement of the residues separating the entrance and the substrate cavities, which may explain their general preference for MAO-B.

**ChC4** was located inside the cavity interacting with Tyr435, Tyr398, Tyr60, Phe343, Asn83, Arg307, Thr314, and Leu328. Two hydrogen bonds where generated with Asn83 and, via its C-3′ hydroxyl group, Thr316. **ChC2** actually when interact with the amino acids into the pocket adopt a planar conformation because the hydrogen bond confirms our discussion that could be responsible for none activity of **ChC2** in rMAO-B. A quantum geometric optimization of **ChC2** and **ChC4** at the M05-2X-D3/6-31G(d,p) level of theory, showed their C-2′ and C-3′ hydroxyl groups pointing in opposite directions, suggesting different preferred intramolecular interactions (Figure 1). Both the different electronic potential distribution and the resulting preferred intermolecular interaction might be responsible for the difference in in IC_50_ values.

The best three ligands obtained from the docking exhibit low *K_d_* values, these ligands are **ChC4**, **ChC9** and **ChC13**, which means that these ligand/rMAO-B complexes are the most stable in the series. These results are consistent with those obtained in the docking experiments in which these complexes were the most stable according to their ∆*E_binding_* values. The proposed tolerable values of *LE* for inhibitor candidates are *LE* > 0.3 kcal·mol^−1^ [34,35,36]. According to this reference value, **ChC4** is a good prospect for development as an rMAO-B inhibitor with a *LE* value of 0.408. Although **ChC2** and **ChC4** have similar *K_d_*, *LE* and ∆*E_binding_* values, the **ChC2** molecule does not show in vitro activity against rMAO-B, on the other hand, ChC4 has a good inhibitory activity against rMAO-B, since micromolar concentrations are needed to inhibit it, which is consistent with the values obtained for the *K_d_*. ChC1 would appear to be almost as good, with *LE* = 0.404, but again its activity, if any, is worse than our cutoff value. The low micromolar-active **ChC5**, **ChC6**, **ChC9** and **ChC11** have *LE* values of 0.384, 0.372, 0.380 and 0.356, respectively, and the less (or in-) active **ChC13** and **ChC14** have *LE* values of 0.380 and 0.391.

### 2.4. Analysis of Molecular Dynamics Simulations

Molecular dynamics simulations were performed for 100 ns to analyze the conformational stability of the rMAO-B/**ChC2** and rMAO-B/**ChC4** complexes. The RMSD, a quantitative parameter, was used to estimate the stability of the protein-ligand systems and the apoprotein. The RMSD in Figure 2A shows that the rMAO-B/**ChC2** and rMAO-B/**ChC4** complexes remain highly stable throughout the simulation time. We can see that the structures of the complexes does not change significantly. The RMSD values for the **ChC4** complex are remarkably constant about 1.5 Å, with a very slight instability and increase near the end of the simulation. The **ChC2** complex shows similar, somewhat less stable behavior for almost 40 ns, and then its RMSD value falls abruptly to about 1.0 Å and rises slowly with appreciable fluctuations to about 1.2 Å at 100 ns, indicative of weaker binding in the rMAO-B site. However, a maximum difference of 3.0 Å in the RMSD is taken to indicate that a system is in equilibrium [37], so this condition is fulfilled by both compounds. To complement the analysis carried out calculating the RMSD, the Radius of Gyration (R_Gyr_) was analyzed for the same runs. From this analysis (Figure 2B), we can conclude that the R_Gyr_ of **ChC2** and **ChC4** oscillate in a narrow interval between 4.3–4.8 Å. These stable values during the 100 ns simulation indicate again that ligand binding does not induce major conformational changes in the protein structure.

Structural studies in MAO have shown that two residues Tyr398 and Tyr435 in MAO-B located in the active site approximately perpendicular to the FAD play a functional role in this enzyme, acting as a cofactor stabilizing the active site, forming an aromatic box whose function is to stabilize the ligand [13]. Molecular simulation results show a difference in the interaction of the compounds **ChC2** and **ChC4** with the FAD cofactor, see Figure 3A. Compound **ChC2** shows a spacing that fluctuates between 17.0 Å and 20.0 Å from its original position, signifying a null interaction with the FAD cofactor. On the other hand, the compound **ChC4** is within the range of interaction with the FAD cofactor. This distance was measured between the nitrogen atom of the alloxazine planar ring of FAD and the center of the benzaldehyde aromatic ring of compounds, Figure 3B.

Molecular dynamics simulations showed of rMAO-B that residues that interact with the ligands ChC2 and ChC4, see Figure 4. The most frequent residues in rMAO-B/**ChC2** were Ile164, Ile199, Leu167, Leu171, Phe168, Pro104, Trp119, Val316, Phe103, Pro102, Tyr115 and Thr196. In contrast, the most frequent residues in rMAO-B/**ChC4** were Ile164, Ile199, Leu171, Phe168, Pro104, Trp119, Tyr326, Val316, Cys172 and Tyr115 with van der Waals and hydrogen bonds interactions. Highlighting residues Cys172 and Tyr326, which are important for the active site of the rMAO-B flavoprotein. Tyr326 and Cys172 are key residues that determines substrate and inhibitor specificity, also exhibits conformational changes on the inhibitor binding and restricts the binding of certain inhibitors (e.g., harmine) to human MAO-B [38]. These results documents that ChC4 is a reversible inhibitor of rMAO-B.

The analyses of trajectories indicate that during most of the simulation the ligand **ChC4** maintain hydrogen bonds with residues of the active site of rMAO-B. However, the number of hydrogen bonds formed was different for **ChC2** and **ChC4** (Figure 5). **ChC2** formed two hydrogen bonds between the residues Glu483 and Tyr115, highlighting the participation of the residues Val316, Ala325, Ile164 and Leu167. Finally, **ChC4** formed two hydrogen bonds with the Phe168, Cys172, Ile164 and Tyr115, highlighting the participation of the residues Ile199, Trp119 and Tyr326. These residues, see Figure 6, are consistent with previous theoretical-experimental studies carried out [39,40] where they detail the interaction that some of the synthesized compounds have with the active site of the rMAO-B. This difference in the formation of hydrogen bonds with key residues in rMAO-B could be explained the difference in experimental activity between the **ChC2** and **ChC4** compounds.

Finally, the binding free energy (MM/GBSA) was computed after the MD simulation; the last 70 ns for all the complexes and the results are given in Table 4. The compound **ChC2** has a binding free energy of -29.06 kcal·mol^−1^ with rMAO-B enzyme, while the compound **ChC4** showed relatively binding free energy of –25.87 kcal·mol^−1^. The results obtained from MM/GBSA show a slight difference in their binding free energy between **ChC2** and **ChC4** compounds bound to rMAO-B. This slight difference is due to the R1 to R2 position of the hydroxyl group in benzaldehyde aromatic ring. In particular, the **ChC4** compound has a better activity at the experimental and in silico level.

### 2.5. In Silico Pharmacokinetic Prediction

A good drug candidate is absorbed in required time and well distributed throughout the system for its effective metabolism and action. Toxicity is another very important factor that often overshadows the ADME behaviour. SwissADME explorer online was used for in silico prediction of drug likeness of the synthesized compounds (**ChC1**–**ChC14**) based on various molecular descriptors and the results are presented in Table 5.

The most potent compound ChC4 in biological experiment data having logP value of 2.97, it’s clear that it doesn’t violate of five Lipinski rules, while the other molecules have logP values in the range of 2.90–4.48 and are expected to be orally active. In addition, the logS values for **ChC4** have a value of −4.94 indicating proper solubility, which is an indication of favorable drug like property, makes compound **ChC4** promising drug candidate for further research and development. Thirteen of fourteen synthesized molecules do not break the rules of Lipinski, Ghose, Veber, Egan and Muegge, since the molecule **ChC12** breaks the rules of Ghose and Muegge.

The Boiled-egg model is proposed as an accurate predictive model that works by computing the lipophilicity and polarity of small molecules. The Boiled-egg analysis of the fourteen molecules (Figure 7) has shown that compounds **ChC1**, **ChC3**, **ChC5**, **ChC6**, **ChC12**, **ChC13** and **ChC14** are highly absorbable at the brain barrier, whereas compounds **ChC2**, **ChC4**, **ChC7**, **ChC8**, **ChC9**, **ChC10** and ChC11 are highly absorbable in the gastrointestinal tract.

The ADMET properties showed much similarity among the thirteen molecules that can be used for advanced clinical trials.

## 3. Materials and Methods

### 3.1. Solvents and Reagents

Solvents and reagents (analytical grade and spectroscopic grade) were obtained from Sigma-Aldrich (St. Louis, MO, USA) and Merck (Darmstadt, Germany). Melting points were determined on a Galen III hot-plate microscope (Reichert-Jung, St. Louis, MO, USA) equipped with a thermocouple. ^1^H- and ^13^C-NMR spectra were recorded on a 400 MHz multidimensional spectrometer (Bruker Corporation, Billerica, MA, USA) using the solvent or the TMS signal as an internal standard.

### 3.2. Synthesis

*3-Cinnamoyl-7-methoxy-2H-chromen-2-one* (**ChC1**). 3-Acetyl-7-methoxy-2H-chromen-2-one (0.44 g, 2.0 mmol) and benzaldehyde (0.21 g, 2.0 mmol) were dissolved in 25 mL of DCM and to this solution 0.5 mL of piperidine were added. The mixture was kept at reflux temperature, monitoring the reaction by TLC for 10 h. The solution was concentrated under reduced pressure and dissolved in a small aliquot of DCM and then MeOH was added in excess to induce precipitation. This procedure was performed twice. The precipitate was finally purified by column chromatography on silica gel eluting with DCM: 0.25 g yellow solid, 40.8%, m.p. 190–192 ºC; ^1^H NMR (CDCl_3_): δ 8.59 (s, 1H, Ar-H), 8.01 (d, 1H, *J* = 15.8 Hz, Ar-CH), 7.85 (d, 1H, *J* = 15.8 Hz, CO-CH=), 7.67 (s, 2H, Ar-H), 7.56 (d, 1H, *J* = 8.6 Hz, Ar-H), 7.40 (s, 3H, Ar-H,), 6.90 (dd, 1H, *J* = 8.6, 1.0 Hz, Ar-H), 6.85 (s, 1H, Ar-H), 3.91 (s, 3H, OCH_3_). ^13^C-NMR (CDCl_3_): δ 56.1, 100.2, 112.4, 113.8, 124.0, 129.0, 130.6, 131.5, 135.3, 144.3, 148.5, 157.8, 160.0, 165.0, 186.3.

*(E)-3-(3-(2-Hydroxyphenyl)acryloyl)-7-methoxy-2H-chromen-2-one* (**ChC2**). 3-Acetyl-7-methoxy-2*H*-chromen-2-one (0.44 g, 2.0 mmol) and *o*-hydroxybenzaldehyde (0.24 g, 2.0 mmol) were reacted and worked up according to the previous procedure: 1.75 g, pale white solid, 95.6%, m.p.; 188–190 °C. ^1^H-NMR (DMSO-d_6_), δ 10.43 (s, 1H, OH), 8.71 (s, 1H, Ar-H), 8.04 (d, 1H, *J* = 15.9 Hz, Ar-CH=), 7.92 (d, 1H, *J* = 8.8 Hz, Ar-H), 7.89 (d, 1H, *J* = 15.9 Hz), 7.69 (dd, 1H, *J* = 7.7, 1.0 Hz, Ar-H), 7.35 (t, 1H, *J* = 7.0 Hz, Ar-H), 7.13 (d, 1H, *J* = 2.2 Hz, Ar-H), 7.08 (dd, 1H, *J* = 8.6, 2.2 Hz, Ar-H), 7.00 (d, 1H, *J* = 8.0 Hz, Ar-H), 6.94 (t, 1H, *J* = 7.5 Hz, Ar-H), 3.96 (s, 3H, OCH_3_), ^13^C-NMR (DMSO-*d_6_*): δ 56.7, 100.8, 112.5, 113.9, 116.8, 120.0, 121.8, 121.9, 124.3, 129.2, 132.3, 132.6, 139.5, 148.2, 157.4, 157.8, 159.4, 165.1, 187.1.

*(E)-7-methoxy-3-(3-(2-methoxyphenyl)acryloyl)-2H-chromen-2-one* (**ChC3**). 3-Acetyl-7-methoxy-2*H*-chromen-2-one (0.44 g, 2.0 mmol) and *o*-methoxybenzaldehide (0.27 g, 2.0 mmol), were reacted and worked up according to the previous procedure: 0.330 g, pale white solid, 49%, m.p.. 184–186 °C; ^1^H-NMR (CDCl_3_): δ 8.55 (s, 1H, =C-H), 8.21 (d, 1H, Ar-CH=, *J* = 15.8 Hz), 8.04 (d, 1H, CO-CH=, *J* = 15.8 Hz), 7.71 (d, 1H, Ar-H, *J* = 7.6 Hz), 7.56 (d, 1H, Ar-H, *J* = 8.4 Hz), 7.37 (t, 1H, Ar-H, *J* = 7.9), 6.98 (t, 1H, Ar-H, *J* = 7.7 Hz), 6.92 (d, 1H, Ar-H, *J* = 8.2 Hz), 6.90 (d, 1H, Ar-H, *J* = 8.5 Hz), 6.85 (s, 1H, Ar-H), 3.91 (s, 6H, 2 OCH_3_); ^13^C-NMR (DMSO-d_6_): δ 55.6, 56.0, 100.4, 111.2, 112.5, 113.7, 120.8, 121.9, 124.1, 124.6, 129.3, 131.3, 132.0, 139.9, 148.2, 157.6, 159.0, 159.8, 165.0, 186.8.

*(E)-3-(3-(3-hydroxyphenyl)acryloyl)-7-methoxy-2H-chromen-2-one* (**ChC4**). 3-Acetyl-7-methoxy-2*H*-chromen-2-one (0.44 g, 2.0 mmol) and *m*-hydroxybenzaldehyde (0.24 g, 2.0 mmol) were reacted and worked up according to the previous procedure: 0.195 g, white solid, 30%, m.p. 184–186 °C; ^1^H NMR (DMSO-d_6_): δ 9.77 (sbr, 1H), 8.76 (s, 1H, =C-H), 7.94 (d, 1H, Ar-H, *J* = 8.0 Hz), 7.82 (d, 1H, Ar-CH=, *J* = 15.9 Hz), 7.71 (d, 1H, CO-CH=, *J* = 15.9 Hz), 7.62 (d, 1H, Ar-H, *J* = 8.6 Hz), 7.33 (m, 1H, Ar-H), 7.25-7.17 (m, 2H, Ar-H), 7.15 (s, 1H, Ar-H), 7.09 (dd, 1H, Ar-H, *J* = 8.0, 1.0 Hz), 6.94 (dd, 1H, Ar-H, *J* = 8.0, 1.0 Hz), 3.97 (s, 3H, OCH_3_); ^13^C NMR (DMSO-d_6_): 56.7, 100.9, 112.5, 114.0, 114.8, 118.5, 120.5, 121.6, 124.9, 130.6, 132.5, 136.3, 143.9, 148.6, 157.5, 158.2, 159.4, 165.3, 186.7.

*(E)-7-methoxy-3-(3-(3-methoxyphenyl)acryloyl)-2H-chromen-2-one* (**ChC5**). 3-Acetyl-7-methoxy-2*H*-chromen-2-one (0.44 g, 2.0 mmol) and *m*-methoxybenzaldehide (0,27 g, 2.0 mmol), were reacted and worked up according to the previous procedure: 0,280 g, pale white solid, 42%, m.p. 164–166 °C; ^1^H-NMR (CDCl_3_): δ 8.58 (s, 1H, =C-H), 7.99 (d, 1H, Ar-CH=, *J* = 15.9 Hz), 7.82 (d, 1H, CO-CH=, *J* = 15.9 Hz), 7.57 (d, 1H, Ar-H, *J* = 8.0 Hz), 7.35-7.25 (m, 2H, Ar-H), 7.18 (s, 1H, Ar-H) 6.96 (dd, 1H, Ar-H, *J* = 8.0, 2.0 Hz), 6.91 (dd, 1H, Ar-H, *J* = 8.8, 2.0 Hz), 6.85 (d, 1H, Ar-H, *J* = 2.0 Hz), 3.83 (s, 3H, OCH_3_), 3.76 (s, 3H, OCH_3_); ^13^C-NMR (CDCl_3_): δ 55.8, 56.5, 100.8, 112.8, 113.9, 114.3, 117.1, 121.7, 122.0, 124.9, 130.3, 131.8, 136.8, 144.8, 149.0, 158.1, 160.2, 160.3, 165.8, 186.8.

*(E)-7-methoxy-3-(3-(4-methoxyphenyl)acryloyl)-2H-chromen-2-one* (**ChC6**). 3-Acetyl-7-methoxy-2*H*-chromen-2-one (0.44 g, 2.0 mmol) and *p*-methoxybenzaldehide (0.27 g, 2.0 mmol), were reacted and worked up according to the previous procedure: 0.26 g, pale white solid, 39%, m.p. 158–160 °C; ^1^H-NMR (CDCl_3_): δ 8.58 (s, 1H, =C-H), 7.91 (d, 1H, Ar-CH=, *J* = 15.9 Hz), 7.84 (d, 1H, CO-CH=, *J* = 15.9 Hz), 7.64 (d, 2H, Ar-H, *J* = 8.0 Hz), 7.56 (d, 1H, Ar-H, *J* = 8.8 Hz), 6.93 (d, 2H, Ar-H, *J* = 8.0 Hz), 6.91 (dd, 1H, Ar-H, *J* = 8.8, 2.0 Hz), 6.85 (d, 1H, Ar-H, *J* = 2.0 Hz), 3.92 (s, 3H, OCH_3_), 3.86 (s, 3H, OCH_3_); ^13^C-NMR (CDCl_3_): δ 57.2, 57.8, 102.0, 114.2, 115.5, 116.1, 123.4, 123.6, 129.6, 132.5, 133.0, 146.3, 150.0, 159.4, 161.6, 163.5, 166.8, 188.0.

*(E)-3-(3-(3,4-dimethoxyphenyl)acryloyl)-7-methoxy-2H-chromen-2-one* (**ChC7**). 3-Acetyl-7-methoxy-2*H*-chromen-2-one (0.44 g, 2.0 mmol) and 3,4-dimethoxybenzaldehide (0.33 g, 2.0 mmol) were reacted and worked up according to the previous procedure: 0.42 g, bright yellow solid, 57%, m.p. 182–184 °C; ^1^H-NMR (CDCl_3_): δ 8.55 (s, 1H, =C-H), 8.16 (d, 1H, Ar-CH=, *J* = 15.9 Hz), 8.00 (d, 1H, CO-CH=, *J* = 15.9 Hz), 7.55 (d, 1H, Ar-H, *J* = 8.6 Hz), 7.34 (d, 1H, Ar-H, *J* = 1.0 Hz), 7.19 (dd, 1H, AR-H, *J* = 8.0, 8.0 Hz) 7.07 (d, 1H, Ar-H, *J* = 8.0 Hz), 7.00 (dd, 1H, Ar-H, *J* = 8.0, 2.0 Hz), 6.94(d, 1H, Ar-H, *J* = 2.0 Hz), 3.90 (s, 3H, OCH_3_), 3.89 (s, 3H, OCH_3_), 3.87 (s, 3H, OCH_3_); ^13^C-NMR (CDCl_3_): δ 55.9, 56.0, 61.5, 100.4, 112.4, 113.8, 114.4, 119.9, 121.6, 124.2, 125.5, 129.2, 131.3, 139.1, 148.4, 149.2, 153.2, 157.7, 159.7, 165.1, 186.7.

*(E)-3-(3-(2,5-dimethoxyphenyl)acryloyl)-7-methoxy-2H-chromen-2-one* (**ChC8**). 3-Acetyl-7-methoxy-2*H*-chromen-2-one (0.44 g, 2.0 mmol) and 2,5-dimethoxybenzaldehide (0.33 g, 2.0 mmol), were reacted and worked up according to the previous procedure: 0,45 g, bright yellow solid, 62%, m.p. 174–176 °C; ^1^H-NMR (CDCl_3_): δ 8.55 (s, 1H, =C-H), 8.17 (d, 1H, Ar-CH=, *J* = 15.6 Hz), 8.0 (d, 1H, CO-CH=, *J* = 15.6 Hz), 7.6 (dd, 1H, Ar-H, *J* = 8.0, 9.0 Hz), 7.2 (d, 1H, Ar-H, *J* = 2.0 Hz), 6.92-6.97 (m, 4H, Ar-H), 3.91 (s, 3H, OCH_3_), 3.87 (s, 3H, OCH_3_), 3.82 (s, 3H, OCH_3_); ^13^C-NMR (CDCl3): δ 56.2, 56.4, 56.6, 100.7, 100.7, 112.8, 112.9, 113.7, 114.1, 114.3, 118.3, 125.1, 131.7, 131.9, 139.9, 148.6, 153.9, 154.0, 158.0, 165.4, 187.1.

*(E)-3-(3-(benzo[d][1,3]dioxol-5-yl)acryloyl)-7-methoxy-2H-chromen-2-one* (**ChC9**). 3-Acetyl-7-methoxy-2*H*-chromen-2-one (0.44 g, 2.0 mmol) and piperonal (0.30 g, 2.0 mmol) were reacted and worked up according to the previous procedure: 0.29 g, yellow solid, 41%, m.p. 178–180 °C; ^1^H-NMR (DMSO-d_6_): δ 8.62 (s, 1H, =C-H), 7.84 (d, 1H, *J* = 12.0 Hz), 7.63 (d, 1H, Ar-CH=, *J* = 15.7 Hz), 7.56 (d, 1H, CO-CH= *J* = 15.7 Hz), 7.68 (d, 1H, Ar-H, *J* = 8.0 Hz), 7.33 (s, 1H, Ar-H), 7.25 (d, 1H, Ar-H, *J* = 8.0 Hz), 7.06 (s, 1H, Ar-H), 7.03-6.94 (m, 3H, Ar-H), 6.07 (s, 2H, OCH_2_O), 3.87 (s, 3H, OCH_3_); ^13^C-NMR (CDCl_3_): δ 55.8, 100.6, 101.2, 106.7, 108.4, 111.5, 122.8, 125.7, 128.1, 130.8, 134.6, 142.2, 147.3, 147.9, 153.6, 160.9, 163.5, 187.1.

*(E)-3-(3-(4-hydroxy-3-methoxyphenylacryloyl)-7-methoxy-2H-chromen-2-one* (**ChC10**). 3-Acetyl-7-methoxy-2*H*-chromen-2-one (0.44 g, 2.0 mmol) and vainillin (0.30 g, 2.0 mmol), were reacted and worked up according to the previous procedure: 0,340 g, yellow solid, 48.3%, m.p. 210–212 °C; ^1^H-NMR (CDCl_3_): δ 8.65 (s, 1H, =C-H), 8.27 (d, 1H, Ar-CH=, *J* = 16 Hz), 8.10 (d, 1H, CO-CH=, *J* = 16 Hz), 7.65 (d, 1H, Ar-H, *J* = 8.0 Hz), 7.32 (d, 1H, Ar-H, *J* = 2.0 Hz), 7.45 (dd, 1H, Ar-H, *J* = 8.8, 2.0 Hz) 7.00 (dd, 1H, Ar-H, *J* = 8.8, 2.0 Hz), 6.91 (d, 1H, Ar-H, *J* = 8.8 Hz), 6.93 (d, 1H, Ar-H, *J* = 2.0 Hz), 3.97 (s, 3H, OCH_3_), 3.92 (s, 3H, OCH_3_); ^13^C-NMR (CDCl_3_): δ 55.9, 56.0, 100.4, 112.4, 112.5, 113.3, 113.7, 117.9, 121.7, 124.6, 124.8, 131.3, 139.6, 148.2, 153.5, 153.6, 157.6, 159.7, 165.7, 186.7.

*(E)-3-(3-(4-mercaptophenyl)acryloyl)-7-methoxy-2H-chromen-2-one* (**ChC11**). 3-Acetyl-7-methoxy-2*H*-chromen-2-one (0.44 g, 2.0 mmol) and methyl(4-vinylphenyl)sulfane (0.27g, 2.0 mmol), were reacted and worked up according to the previous procedure: 0,305 g, yellow solid, 43%.m.p. 196–198 °C; ^1^H-NMR (CDCl_3_): δ 8.58 (s, 1H, =C-H), 7.99 (d, 1H, Ar-CH=, *J* = 15.7 Hz), 7.82 (d, 1H, CO-CH=, *J* = 15.7 Hz), 7.58 (m, 3H, Ar-H), 7.26 (m, 3H, Ar-H 6.91 (dd, 1H, Ar-H, *J* = (8.0, 2.0), 6.86 (d, 1H, Ar-H, *J* = 2.0 Hz), 3.91 (s, 3H, OCH_3_), 2.50 (s, 3H, SCH_3_); ^13^C-NMR (CDCl_3_): δ 15.6, 56.4, 100.8, 112.9, 114.3, 121.9, 123.6, 126.3, 129.6, 131.7, 131.9, 142.9, 144.4, 148.8, 158.1, 160.2, 165.5, 165.6, 186.7.

*(E)-3-(3-(3,5-dibromo-4-methoxyphenyl)acryloyl)-7-methoxy-2H-chromen-2-one* (**ChC12**). 3-Acetyl-7-methoxy-2*H*-chromen-2-one (0,44 g, 2.0 mmol) and 3,5-dibromo-4-methoxybenzaldehyde (0.58 g, 2.0 mmol), were reacted and worked up according to the previous procedure: 0.21 g, yellow solid, 21.4%, m.p. 228–230 °C; ^1^H-NMR (CDCl_3_): δ 8.57 (s, 1H, =C-H), 8.18. (s, 2H, Ar-H), 7.95 (d, 1H, Ar-CH=, *J* = 16 Hz), 7.81 (d, 1H, CO-CH=, *J* = 16 Hz), 7.63 (d, 1H, Ar-H, *J* = 8.0 Hz), 7.22 (dd, 1H, Ar-H, *J* = 8.0, 2.0 Hz), 6.98 (d, 1H, Ar-H, *J* = 2.0 Hz) 4.01 (s, 3H, OCH_3_), 3.95 (s, 3H, OCH_3_); ^13^C-NMR (CDCl_3_): δ 60.9, 56.5, 100.5, 110.8, 111.7, 117.4, 123.7, 126.9, 132.3, 133.6, 136.1, 141.9, 147.7, 154.0, 154.5, 157.6, 159.4, 163.2, 188.1.

*(E)-3-(3-(4-(dimethylamino)phenyl)acryloyl)-7-methoxy-2H-chromen-2-one* (**ChC13**). 3-Acetyl-7-methoxy-2*H*-chromen-2-one (0.44 g, 2.0 mmol) and 4-(dimethylamino)benzaldehyde (0.27 g, 2.0 mmol), were reacted and worked up according to the previous procedure: 0.15 g, red solid, 22%, m.p. 220–222 °C; ^1^H-NMR (CDCl_3_): δ 8.56 (s, 1H, =C-H), 7.97 (d, 1H, Ar-CH=, *J* = 15.7 Hz), 7.81 (d, 1H, CO-CH=, *J* = 15.7 Hz), 7.58 (d, 1H, Ar-H, *J* = 8.8 Hz), 7.54 (d, 2H, Ar-H, *J* = 8.6 Hz), 6.88 (d, 1H, Ar-H, *J* = 8.0, 2.0 Hz), 6.85 (d, 1H, Ar-H, *J* = 2.0 Hz), 6.68 (d, 2H, Ar-H, *J* = 8.8 Hz), 3.90 (s, 3H, OCH_3_), 3.04 (s, 6H, N(CH_3_)_2_); ^13^C-NMR (CDCl_3_): δ 40.5, 56.4, 64.1, 100.7, 112.2, 113.0, 114.0, 119.3, 122.6, 123.3, 131.4, 131.5, 146.4, 148.2, 152.6, 158.0, 160.3, 165.2, 186.4.

*(E)-3-(3-(4-bromophenyl)acryloyl)-7-methoxy-2H-chromen-2-one* (**ChC14**). 3-Acetyl-7-methoxy-2*H*-chromen-2-one (0.44 g, 2.0 mmol) and *p*-methoxybenzaldehide (0.27 g, 2.0 mmol), were reacted and worked up according to the previous procedure: 0.27 g, pale white solid, 35%, m.p. 158–160 °C; ^1^H-NMR (CDCl_3_): δ 8.76 (s, 1H, =C-H), 7.94 (d, 1H, Ar-H, *J* = 8.0, 8.0 Hz), 7.82 (d, 1H, Ar-CH=, *J* = 16 Hz), 7.71 (d, 1H, Ar-CH=, *J* = 16 Hz), 7.33 (d, 1, Ar-H, *J* = 8.0, 8.0 Hz), 7.25-7.18 (m, 2H, Ar-H), 7.15 (s, 1H, Ar-H), 7.09 (dd, 1H Ar-H, *J* = 8.0, 2.0 Hz), 6.94 (dd, 1H Ar-H, *J* = 8.0, 2.0 Hz), 3.97 (s, 3H, OCH_3_); ^13^C-NMR (CDCl_3_): δ 57.2, 60.8, 110.2, 112.4, 118.5, 126.1, 129.6, 133.0, 134.2, 136.5, 142.0, 147.3, 154.1, 154.4, 159.6, 160.5, 186.7.

### 3.3. Biological Assessment

The effect of coumarin derivatives on MAO-A and MAO-B were measured using a suspension of crude rat brain mitochondria as enzyme source. 4-Dimethylaminophenethylamine (4-DMAPEA, 2.5 µM) and 5-hydroxytryptamine (5-HT, 100 mM) were used as substrates selective of MAO-B or MAO-A, respectively. Evaluation of the test compounds on rMAO activity was executed by measuring their effects on the production of 4-dimethylaminophenylacetic acid (DMAPAA) by rMAO-B and 5-hydroxyindoleacetic acid (5-HIAA) by rMAO-A with O_2_ using HPLC-ED (L-7110 LaChrom and amperometric detector L-3500 LaChrom Recipe, Hitachi, (Tokyo, Japan) (for more detail see methodological references [41,42]). The IC_50_ values (average ± SD was measured in two independent experiments each in triplicate) were assessed representing percentage of inhibition in function of the negative logarithm of different inhibitor concentrations (10^−4^ to 10^−8^) using the GraphPad Prism software [43].

### 3.4. Computational Analysis

#### 3.4.1. Homology Modeling

Human monoamine oxidase B (hMAO-B) at 1.6Å resolution was used as template (PDB code 1OJ9) to obtain a 3D structure of rat MAO-B (rMAO-B) using homology modeling. The amino acid sequence and crystal structure of the protein was extracted from NCBI and PDB databases [44,45] considering the high level of amino acid identity (around 90%) the target protein and template were aligned through a single alignment using MultAlin interface [46]. MODELLER9v6 program [47] was used and 100 structures were prepared using standard parameters and the outcomes were ranked on the basis of the internal scoring function of the program (DOPE score). The best model was chosen as the target model. The cofactor FAD was placed inside of MAO using the corresponding crystal coordinates. To analyze the rMAO-B model, VMD program [48] was used to evaluate the 3D distribution and general physical chemistry characteristics. Then, stereochemical and energetic quality of the homology models was evaluated using PROSAII server [49], ANOLEA server [50] and Procheck program [51]. The crystal structure of rMAO-A (PDB code 1O5W [52]) and model of rMAO-B isoform were submited to H++ server [53,54] to computes pK values of ionizable groups and adds missing hydrogen atoms according to the specified pH of the environment as is described in H++ server.

#### 3.4.2. Molecular Docking

Coumarin-chalcone hybrids were docked in the binding cavity of rMAO-A (PDB code 1O5W) and homology model for rMAO-B using AutoDock 4.012 suite. In general, the grid maps were calculated using the AutoGrid 4.0 option and were centered on the sites described before. The volume chosen for the grid maps were made up of 60 × 60 × 60 points, with a grid-point spacing of 0.375 Å. The author’s option was used to define the rotating bond in the ligand. In the Lamarckian genetic algorithm (LGA) dockings, an initial population of random individuals with a population size of 150 individuals, a maximum number of 2.5 × 10^7^ energy evaluations, a maximum number generation of 27,000, a mutation rate of 0.02 and crossover rate of 0.80 were employed. Each complex was built using the lowest docked-energy binding positions. Van der Waals interaction cutoff distances were set at 12 Å and dielectric constant was 10. The partial charges of each ligand were determined with PM6-D3H4 semi-empirical method [55,56] implemented in the MOPAC2016 [57] software. PM6-D3H4 [56] introduces dispersion and hydrogen-bonded corrections to the PM6 method.

#### 3.4.3. Ligand Efficiency Approach

Ligand efficiency (LE) calculations were performed using one parameter *K_d_*. The *K_d_* parameter corresponds to the dissociation constant between a ligand/protein, and their value indicates the bond strength between the ligand/protein [34,35,36]. Low values indicate strong binding of the molecule to the protein. *K_d_* calculations were done using the following Equations (1) and (2):(1)$$\Delta G^{0}=-2.303RTlog\left(K_{d}\right)$$
(2)$$K_{d}=10^{\frac{\Delta G^{0}}{2.303RT}}$$
where ∆*G*^0^ is the binding energy (kcal·mol^−1^) obtained from docking experiments, *R* is the gas constant, and *T* is the temperature in Kelvin. In standard conditions of aqueous solution at 298.15 K, neutral pH and remaining concentrations of 1 M. The LE allows us to compare molecules according to their average binding energy [36,58]. Thus, it determined as the ratio of binding energy per non-hydrogen atom, as follows (Equation (3)) [34,35,36,59]:(3)$$LLE=-\frac{2.303RT}{HAC}\log\left(K_{d}\right)$$
where *K_d_* is obtained from Equation (2) and HAC denotes the heavy atom count (i.e., number of non-hydrogen atoms) in a ligand.

#### 3.4.4. Molecular Dynamic Simulations

Two complexes were built for each modeled **ChC2**/rMAO-B and **ChC4**/rMAO-B, and each model was confined inside a periodic simulation box. Water model TIP3P [60] with 20.459 molecules was used as solvent. Furthermore, Na^+^ and Cl^−^ ions were added to neutralize the systems and maintain an ionic concentration of 0.15 mol·L^−1^.The full geometry optimizations of the two molecules were carried out with the density functional theory method by a M05-2X [61]-D3 [62] in conjunction with the 6-31G(d,p) basis set. **ChC2**, **ChC4** and FAD compounds were parametrized using LigParGen web server and implementing the OPLS-AA/1.14*CM1A(-LBCC) force field parameters for organic ligands [63,64,65]. The partial charges of each ligand were determined with generated by the restrained electrostatic potential (RESP) model [66]. MD simulations were carried out using the modeled CHARMM22 and CHARMM36 force fields [67,68] within the NAMD software [69]. First, each system included 20,000 steps of conjugate-gradient energy minimization followed by 10 ns of simulation with the protein backbone atoms fixed and gradually releasing the backbone over 50,000 ps with 10 to 0.001 kcal·mol^−1^Å^−2^ restraints. The total duration of simulation was approximately 100 ns for each system. During the MD simulations, motion equations were integrated with a 1 fs time step in the NPT ensemble at a pressure of 1 atm. The SHAKE algorithm was applied to all hydrogen atoms, and the van der Waals cutoff was set to 12 Å. The temperature was maintained at 310 K, employing the Nosée-Hoover thermostat method with a relaxation time of 1 ps. The Nosée-Hoover-Langevin piston was used to control the pressure at 1 atm. Long-range electrostatic forces were taken into account by means of the particle-mesh Ewald approach. Data were collected every 1 ps during the MD runs. Molecular visualization of the systems and MD trajectory analysis were carried out with the VMD software package [48].

#### 3.4.5. Free Energy Calculation

The molecular MM/GBSA method was employed to estimate the binding free energy of the rMAO-B/ligand complexes. For calculations from a total of 100 ns of MD, the last 70 ns were extracted for analysis, and the explicit water molecules and ions were removed. The MM/GBSA analysis was performed on three subsets of each system: the protein alone, the ligand alone, and the complex (protein-ligand). For each of these subsets, the total free energy (Δ*G_tot_*) was calculated as follows (Equation (4)):(4)$$\Delta G_{tot}=E_{MM}+G_{solv}-T\Delta S_{conf}$$
where *E_MM_* is the bonded and Lennard–Jones energy terms; *G_solv_* is the polar contribution of solvation energy and non-polar contribution to the solvation energy; T is the temperature; and Δ*S_conf_* corresponds to the conformational entropy [70]. Both *E_MM_* and *G_solv_* were calculated using NAMD software with the generalized Born implicit solvent model [71,72]. Δ*G_tot_* was calculated as a linear function of the solvent-accessible surface area, which was calculated with a probe radius of 1.4 Å [73]. The binding free energy of rMAO-B and ligand complexes ($\Delta G_{bind}$) were calculated by the difference where Δ*S_conf_* values are the averages over the simulation (Equation (5)):(5)$$\Delta G_{bind}=\Delta G_{tot}\left(complex\right)-\Delta G_{tot}\left(protein\right)-\Delta G_{tot}\left(ligand\right)$$

#### 3.4.6. ADMET Prediction

The ADMET properties of a compound deal with its absorption, distribution, metabolism, excretion, and toxicity in and through the human body. ADMET, which constitutes the pharmacokinetic profile of a drug molecule, is very essential in evaluating its pharmacodynamic activities. In this study for all molecules, we have used the SwissADME [74] prediction tool, for in silico physicochemical properties such as molecular hydrogen bond acceptor (*HBA*), hydrogen bond donor (*HBD*), weight (*MW*), topological polar surface area (*TPSA*), rotatable bond count (*RB*), octanol/water partition coefficient (*LogP*), water solubility (*LogS*) and skin permeation (*logKp*). Further the ligands were analyzed for Bioavailability property using Boiled Egg analysis [75].

## 4. Conclusions

Fourteen compounds derived from chalcocoumarins were synthesized and evaluated against monoamine oxidase enzyme isoforms. The experimental results obtained against MAO-A and MAO-B show that the compounds **ChC4**, **ChC5**, **ChC6**, **ChC9** and **ChC11** exhibit MAO-B affinity at micro and sub-micromolar concentrations, in particular **ChC4** which shows an IC_50_ value of 0 0.76 ± 0.08 µM. Where compound **ChC4** is highlighted in molecular modeling, ADMET predictions, docking and MM/GBSA calculations, these results suggest that compound **ChC4** has the appropriate interactions with the active site of rMAO-B. Furthermore, the ADMETox values obtained for the compound **ChC4** indicate adequate solubility in the gastrointestinal tract, which is a favourable indication for it to be a promising drug candidate for further research and development. This compound complies with the interactions described for the active site of rMAO-, fitting into a distance close enough to the nitrogen atom of the aloxazine planar ring of FAD to form an interaction necessary for the inhibition of rMAO-B. These analyses may be important initial steps towards the development of new drugs in the fight against depressive disorder and Parkinson’s disease.

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
