# Peer review of "Coumarin-Chalcone Hybrids as Inhibitors of MAO-B: Biological Activity and In Silico Studies"

_molecules, 2021, doi:10.3390/molecules26092430_

Round 1
Reviewer 1 Report
This article aims to discuss the synthesis of fourteen coumarin- derived compounds modified at C3 with an α, β-unsaturated ketone and their biological evaluation against MAO-B.
The manuscript is written comprehensively enough to be understandable.
The paper stated the purpose, discussion and global implication are clearly stated and consistent with the rest of the manuscript.
The authors clearly described coumarins and monoamine oxidase (MAO) in their introduction but it is recommended to add more details about coumarins and their medical uses, they can use these useful reviews:
- Darry M.A.Oliver, Hemachandra Reddy, Small molecules as therapeutic drugs for Alzheimer's disease, Molecular and Cellular Neuroscience, 96, 2019, (47-62). https://doi.org/10.1016/j.mcn.2019.03.001
- Donglai Lv,Zongtao Hu, Lin Lu, Husheng Lu, and Xiuli Xu,Three-dimensional cell culture: A powerful tool in tumor research and drug discovery. Oncol Lett. 2017, 14(6): 6999–7010. doi: 3892/ol.2017.7134
- Cione E, La Torre C, Cannataro R, Caroleo MC, Plastina P, Gallelli L. Quercetin, Epigallocatechin Gallate, Curcumin, and Resveratrol: From Dietary Sources to Human MicroRNA Modulation. Molecules. 2019, 25(1):63. doi: 10.3390/molecules25010063.
The authors addressed their hypothesis and opinion in a reproducible way, methods are adequately described, the results was presented in a clear way which facilitate in reaching a conclusions.
No plagiarism has been detected.
Writing errors:
195: shows
196: We
References: The authors followed perfectly the journal guidelines.
Author Response
We thank the reviewer. His contributions were very important to improve this manuscrip.
Attached file with responses

Reviewer 2 Report
It was my pleasure to review this research work by Moya-Alvarado et. al. This is a good piece of research work. The presentation of results in the manuscript is very good, and this is a good example of the use of hybrid analogs for the development of the new drug.
I would like to recommend the article could be published in molecules after a minor revision. The authors need to address the below-mentioned queries.
- Line 25: “modified at C3” change to “modified at C3 carbon of coumarin”.
- Line 64: “at positions 3 or 4” change to “at positions 3 or 4 of coumarin”.
- Line 64/65: “A 7-benzyloxy substituent, and sometimes a halogen atom at C-6 are other modifications [19,20].” The sentence is complete.
- Line 70: “a profound effect” on what?
- It is very confusing “line 61-74” and very hard to understand which kind of scaffold, the author was discussing. If the author could provide a figure with molecules being discussed that could clear the doubt for readers of the manuscript.
- Line 72: “he” change to “author” or “name”.
- The author should show the general structure of hybrid (chalcocoumarin) derivatives by highlighting the point of modification for biological activity.
- The author could assign the compounds by number (4, 5, etc.) instead of ChC as it is confused with C-substituted derivatives.
- For scheme 1, yields are missing and for c condition is incomplete.
- The structure of compound 1 should be resorcinol if the author showing conditions a and b in the arrow.
- Compounds 1, 2, and 3 are reported and commercially available. The author could start the synthesis of ChC1–ChC14 from compound 3 directly. Some of the ChC1–ChC14 are reported compounds. The author could move the synthetic scheme of reported compounds to the supporting information.
- ChC1, ChC2, ChC6, ChC11, and ChC14 compounds are reported, the author should provide an exact reference for these compounds. The author could mention that these compounds were prepared as previously reported (references) and spectral data were consistent with previously published data.
- The author should the structure of compound ChC9 separately as the general structure is confusing.
- Table 1, compound Ch10, R4 will be OCH3 instead of OCH.
- Compound ChC13, recheck R3 {C(CH3)2 }; it should be N(CH3)2.
- Line 110-111: “14 derivatives differing in the substitution pattern of the cinnamyl benzene ring were studied in rat MAO-A and B in order to determine their inhibitory activity MAO.” Is activity against MAO?
- Line 112: Change “screeni” to “screening”.
- The author could have screened all (R1-R4) OMe and OH compounds for a better understanding of the effects of OH and OMe.
- ChC6 is less potent than ChC11; The authors need to correct comparison in the line 125-126.
- The author could have tested “F” and “SH” compounds as well.
- The author could have screened few heterocyclic analogs (Cinnamoyl) for a better understanding of the activity.
- Any explanation for choosing the OMe group at the C-7 position of coumarin.
- Any data of cytotoxicity of these compounds are available?
- Line 286: Instead of “CC” write column chromatography.
- For compounds ChC6, ChC9, and ChC14 (Br or two OMe groups) number of protons not matched in the data.
- The Author could provide spectra of all new compounds in the supporting materials.
- For all change “f. 196-198 °C” to “m.p. 196-198 °C”.
- For mmol or g use same decimal (1 mmol or 1.0 mmol).
- Line 295: change “m.p.;” to “m.p.” and “1H NMR” to “1H NMR”.
- For all “DMSO-d6” change to “DMSOd6”.
- Line 303, 312, 321, 330, 339, 349, 365 and 401: remove “y” add “and” (correct similar errors).
- Line 340, 374, 383….: Change “(0,33 g, 2 mmol)” to “(0.33 g, 2 mmol)” and correct similar errors.
- The author should follow the same format for references, for some of the page numbers are missing.
- The author could provide a clearer picture of molecular docking and ligand efficiency analysis and need more explanation (rewrite) for the best compound ChC4. Did the author try to dock other functionality in place of the OH (R2) group and calculate the LE values?
- Geometry (E vs Z) of olefin of cinnamyl moiety has any effect on activity/or binding?
- Line 172: Change “IC50 values” to “IC50 values”.
Author Response
We appreciate your input to improve this manuscript.
Attached file with responses

Reviewer 3 Report
The manuscript reports on Coumarin-chalcone conjugates and their MAO-B biological activity. The paper is biology-oriented and the sections related to the investigated biological activities of compounds read quite well. On the other hand, the chemistry part of the paper seems disregarded and has a number of issues which should be addressed more carefully.
- The authors should clearly state whether the final products obtained are novel and if so, the spectroscopic characterization of those compounds should be extended. If the compounds are not novel, an appropriate reference citations must be provided.
- The novelty of work should be stated more clearly in the introduction section.
- In the section 3.2, the authors report the NMR (both 1H and 13C) data, but no discussion of the spectra, nor an example of the spectra is provided. Given the fact that the NMR spectroscopy was actually the only technique used for the structure elucidation, the discussion of spectra is mandatory. Moreover, an additional figure with the clear numbering system of atoms should be added. Also, all spectra should be provided in supplementary material.
- The NMR data should be double checked for possible mistakes. For instance in some cases the signals assigned as doublets (d) are accompanied with two coupling constant values, which is incorrect. Please check the multiplicity of signals and the related J values. Again, this requires a clear NMR spectra provided in the supplementary file.
- The figure 1 depicting the synthetic pathway should be corrected. In more detail the conditions a and b should be separated and additional step involving the formylation of resorcinol should be added. In its current form the figure shows that the salicylaldehyde was formylated, while in the text there is a statement (probably the correct one) saying that the salicylaldehyde was a product of Vilsmeyer-haack formylation of the resorcinol.
- The authors should provide a clear explanation why did they methylate the hydroxyl group at the 7 position of 3-acetylcoumarin in the step c of the synthetic pathway. What was the reason for such alteration? It does not seem associated with further synthetic step d as few of the final products have the OH groups.
- The melting point data suggest that the purity of compound was high, although this method is often inaccurate and hence an additional set of data confirming the purity of products is required. In particular the microanalysis data (% C, H, and N) is required. Alternatively the HPLC-MS results should be provided and at least briefly discussed.
- The melting point data m.p. in some cases are abbreviated as p.f., which is incorrect and should be changed.
- The 3-substituted coumarins are known for their highly intensive fluorescence. The compounds reported in this paper are definitely fluorescent, but there is not a single word of discussion on the absorption and fluorescence of the compounds. Such simple measurements would greatly improve the structural characterization of the compounds and maybe shed some light on the relationship (if any) between the fluorescence of compound obtained and their MAO-B inhibition ability.
- Large number of coumarin derivatives are known for their problematic aqueous solubility. Please provide a brief information on solubility of the final products in the manuscript. Please comment in relation to the biological activity observed.
- After implementation of the additional data mentioned the conclusions section should be modified accordingly.
Based on the comments above I strongly recommend the major revision of the manuscript.
Author Response

(The authors gave the same response as above.)

Round 2
Reviewer 3 Report
The Authors provided the satisfactory answers and made the appropriate changes in the manuscript. The supplementary file is updated in a proper manner. Accordingly, I recommend the manuscript for publication.